# Feasibility Study and Clinical Impact of Incorporating Breast Tissue Density in High-Risk Breast Cancer Screening Assessment

Alison Rusnak [1], Shawna Morrison [2], Erika Smith [2], Valerie Hastings [2], Kelly Anderson [2], Caitlin Aldridge [2], Sari Zelenietz [2], Karen Reddick [3], Sonia Regnier [3], Ellen Alie [4], Nayaar Islam [5], Rutaaba Fasih [6], Susan Peddle [6], Erin Cordeiro [7], Eva Tomiak [8] and Jean M. Seely [6,*]

1    Inherited Cancer Program, Children's Hospital of Eastern Ontario, 401 Smyth Road, Ottawa, ON K1H 8L1, Canada
2    Regional Genetics Program, Children's Hospital of Eastern Ontario, Ottawa, ON K1H 8L1, Canada
3    The High-Risk OBSP Program Nurse Navigator, The Ottawa Hospital, Ottawa, ON K1H 8L6, Canada
4    Previously High-Risk OBSP Program Screening Manager, The Ottawa Hospital, Ottawa, ON K1H 8L6, Canada
5    School of Epidemiology and Public Health, The Ottawa Hospital, University of Ottawa, Ottawa, ON K1N 6N5, Canada
6    Department of Radiology, The Ottawa Hospital, University of Ottawa, Ottawa, ON K1N 6N5, Canada
7    Department of Surgery, University of Ottawa, Ottawa, ON K1N 6N5, Canada
8    Department of Genetics, Children's Hospital of Eastern Ontario, University of Ottawa, Ottawa, ON K1N 6N5, Canada
*    Correspondence: jeseely@toh.ca

**Abstract:** Breast tissue density (BTD) is known to increase the risk of breast cancer but is not routinely used in the risk assessment of the population-based High-Risk Ontario Breast Screening Program (HROBSP). This prospective, IRB-approved study assessed the feasibility and impact of incorporating breast tissue density (BTD) into the risk assessment of women referred to HROBSP who were not genetic mutation carriers. All consecutive women aged 40–69 years who met criteria for HROBSP assessment and referred to Genetics from 1 December 2020 to 31 July 2021 had their lifetime risk calculated with and without BTD using Tyrer-Cuzick model version 8 (IBISv8) to gauge overall impact. McNemar's test was performed to compare eligibility with and without density. 140 women were referred, and 1 was excluded (*BRCA* gene mutation carrier and automatically eligible). Eight of 139 (5.8%) never had a mammogram, while 17/131 (13%) did not have BTD reported on their mammogram and required radiologist review. Of 131 patients, 22 (16.8%) were clinically impacted by incorporation of BTD: 9/131 (6.9%) became eligible for HROBSP, while 13/131 (9.9%) became ineligible ($p = 0.394$). It was feasible for the Genetics clinic to incorporate BTD for better risk stratification of eligible women. This did not significantly impact the number of eligible women while optimizing the use of high-risk supplemental MRI screening.

**Keywords:** breast screening; high-risk breast screening; dense breasts; supplemental breast screening; breast MRI

**Key Points:**

*1. Breast cancer risk assessment should be performed for all women ≥25–30 years of age to optimize early detection of breast cancer.*

*2. Women considered at high risk of breast cancer ≥40 years old must undergo screening mammography prior to referral to improve risk assessment.*

*3. Incorporating BTD into risk assessment was feasible, did not increase the overall number of women eligible for B-MRI and optimized supplemental B-MRI screening in women with dense breasts.*

## 1. Introduction

Breast tissue density (BTD) decreases mammographic sensitivity by masking underlying cancers. It is also a well-established independent risk factor for breast cancer (BC) [1–4]. Mammographically dense breasts are very common and may contribute more cancer risk than other significant but less common risk factors [3] including obesity [5] and mitochondrial mutations [6,7].

In order to maximally benefit from early detection of breast cancer, international guidelines recommend that risk assessment for all women begin at 25–30 years of age [8,9]. Risk assessment models have been shown to increase their diagnostic accuracy with incorporation of BTD [10]. The High-Risk Ontario Breast Screening Program (HROBSP) is a population-based program for women who have a lifetime risk (LTR) of BC ≥25% or who carry a genetic mutation for BC [11]. Lifetime risk of BC is assessed by Tyrer-Cuzick model version 8 (IBISv8) [12] or Breast and Ovarian Analysis of Disease Incidence of Carrier Estimation Algorithm (BOADICEA) model (incorporated within the CanRisk tool 1 May 2021). Within HROBSP, a woman aged 30–69 years determined to have LTR assessed to be ≥25% is invited to participate in annual mammographic and B-MRI screening.

In the Genetics Clinic, prior to the introduction of the CanRisk model, IBISv8 was the predominant instrument used to assess BC risk for eligible women who had never been diagnosed with BC and who were not known genetic mutation carriers (unaffected). Version 8 incorporates BTD (for woman age 40+) and other personal risk factors along with family history of breast and ovarian cancer and is considered the most reliable model for assessing BC risk [12]. However, BTD was not included by our clinic before the study as it was not available at time of risk assessment. To assess the impact of incorporating BTD into the IBISv8 calculation, our group performed a retrospective review of 156 unaffected, 40–69-year-old women who had already undergone high-risk BC assessment from 1 November 2019 to 31 March 2020. We determined that 93.4% (146/156) had a prior mammogram and calculated that if BTD had been incorporated in the IBISv8, it would have changed the eligibility of 14.4% (21/146) with overall 4% (5/146) fewer women qualifying for annual screening MRI and mammography. Based on this preliminary work demonstrating the importance of including BTD we set out to prospectively determine the feasibility of incorporating this metric for all women undergoing HROBSP assessment at our centre and study the impact on program eligibility for women requiring B-MRI (B-MRI).

## 2. Methods

In this Research Ethics Board approved study from 1 December 2020 to 31 July 2021, genetic counsellors [GCs] performed risk assessments for all women who met Category B criteria for HROBSP screening eligibility (APPENDIX). All patients received one-on-one meeting with a GC. All appointments were virtual (by video or phone) due to the COVID-19 pandemic. Women were excluded from this study if they were under 40 years of age, as incorporation of BTD in IBISv8 is not validated for this age group. Women known to carry a hereditary BC risk gene were automatically eligible for high-risk screening and were excluded.

GCs completed the usual HROBSP assessments using IBISv8 and calculated lifetime risk (to age 80) with and without including BTD. The calculated risk with density included was used for determination of eligibility to the HROBSP program. For the purpose of the study, both numbers were recorded, along with information regarding the length of time it took to access the BTD.

At our centre, most referrals for HROBSP were sent directly to the OBSP nurse navigator (NN) for triage. The NN ensured that the referral met criteria for HROBSP assessment and then forwarded it to the Genetics clinic. For the study, the NN included the report of the most recent available mammogram with the referral. BTD was assessed visually on mammograms in the region and reported using BI-RADS® categories A, B, C, or D [13] (Figure 1). If BTD was not included on this report, the NN contacted the radiologist (JS or SP) to determine the BTD by reviewing the mammogram and/or report so that this could

be included with the referral. In other instances, referrals for HROBSP came directly to the Genetics clinic and when the GC could not access their mammogram reports from the electronic medical record, the study radiologist (JS or SP) was contacted. Patients who reported having a prior mammogram within the province of Ontario could have their reports and/or actual mammogram images accessed through the electronic medical record and/or the picture archiving computer software (PACS) for review by the radiologist. BTD was not obtained for women with mammograms from outside of Ontario and for those who never had a mammogram; their risk assessment was calculated only without incorporating density. The McNemar's test was performed to compare the number of patients who were eligible with density versus the number of patients who were eligible without density; $p < 0.05$ was used to determine significance.

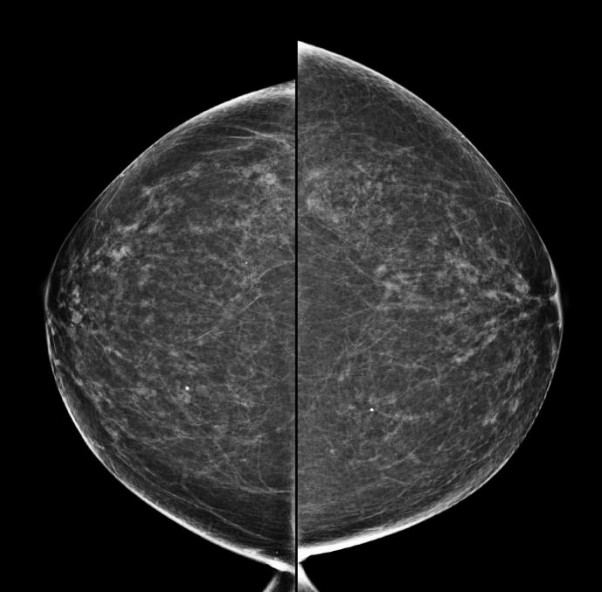

(**A**) ACR BI-RADS category A—fatty replaced breast tissue density.

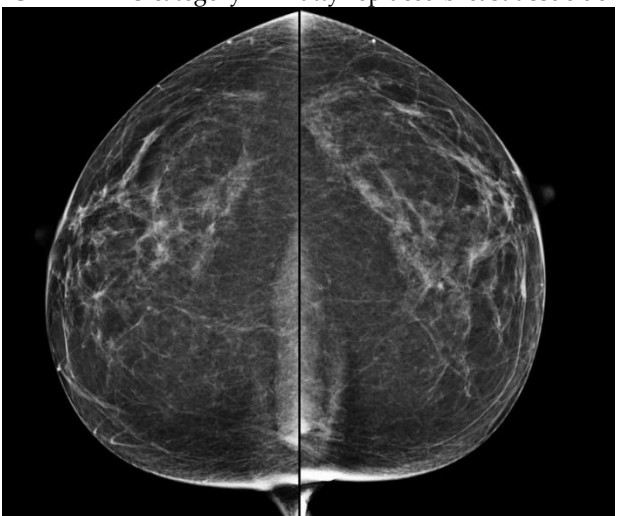

(**B**) ACR BI-RADS category B—scattered breast tissue densities breast tissue density.

**Figure 1.** *Cont.*

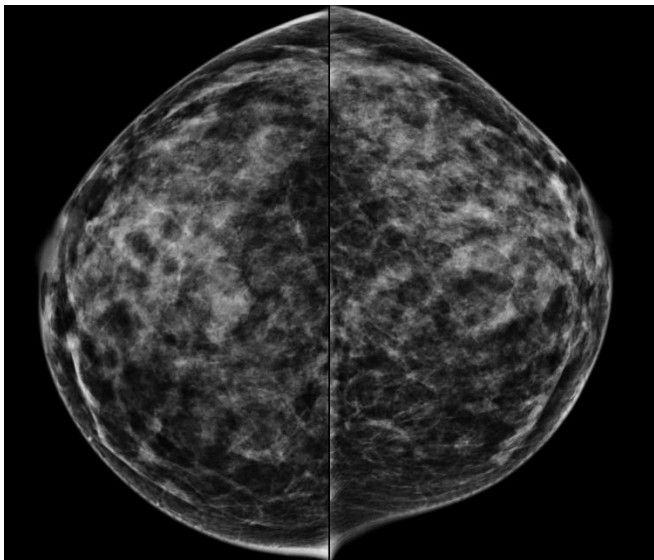

(**C**) ACR BI-RADS category C–heterogeneously dense breast tissue density.

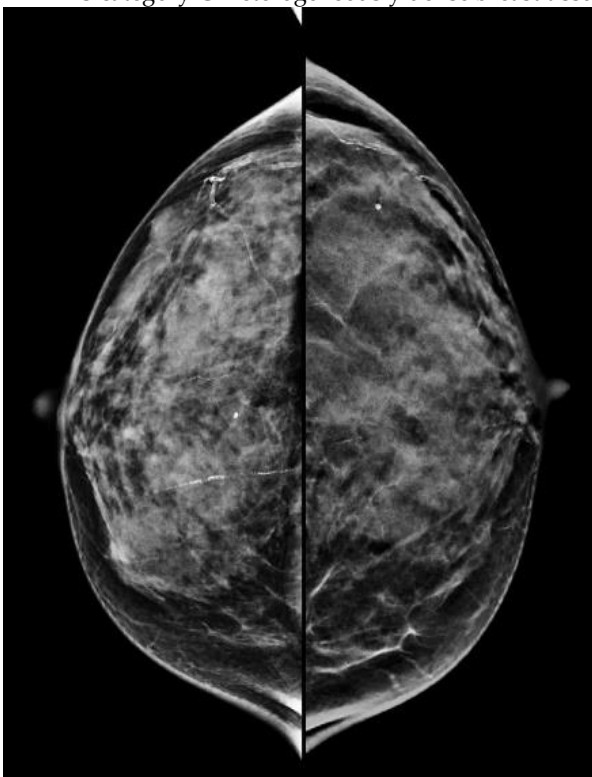

(**D**) ACR BI-RADS category D–extremely dense breast tissue density.

**Figure 1.** ACR BI-RADS categories (**A**–**D**) for craniocaudal mammogram views in 4 different women. As density increases from Categories (**A**) to (**D**), the masking effect increases, and the sensitivity of the mammograms decreases accordingly.

## 3. Results

### 3.1. Impact on Eligibility for B-MRI through the HROBSP Program

During the study interval, 140 women age 40–69 (average 51.4 years) with no prior history of BC underwent HROBSP assessment. One was excluded from this study as she was a known carrier of a *BRCA* gene pathogenic variant and automatically eligible for HR OBSP. Eight of the remaining 139 women (5.8%) never had a mammogram in the past 10 years (7) or only had a mammogram from another country (1) (Table 1).

**Table 1.** The impact on risk assessment and determination of clinical eligibility on addition of breast tissue density according to age groups 40–49 and 50–69.

| Age Groups in Years (Total) | Patient Did Not Have a MG or Was Not Available * | Number of Patients (%) with MG and Density Available | Number of Patients (%) Where Density Increased Calculated Risk ^ | Number of Patients (%) Where Density Decreased Calculated Risk ^^ | Number of Patients (%) Where Density Made Patient Eligible ** | Number of Patients (%) Where Density Made Patient Ineligible | Radiologist Input Required to Assess Density on MG |
|---|---|---|---|---|---|---|---|
| 40–49 (65) | 5 (7.7%) | 60 (92.3%) | 24 (40.0%) | 27 (45.0%) | 6 (10.0%) | 10 (16.7%) | 6 (10.0%) |
| 50–69 (74) | 3 (4.1%) | 71 (95.9%) | 26 (37.1%) | 34 (47.9%) | 3 (4.2%) | 3 (4.2%) | 11 (15.5%) |
| Total (139) | 8 (5.8%) | 131 (94.2%) | 50 (38.5%) | 61 (46.5%) | 9 (6.9%) | 13 (9.9%) | 17 (13.0%) |

MG = mammogram. * Women age 40–49 y had never undergone a mammogram while those 50–69 had mammograms in another province or country and the report was not available. ** eligibility was determined if calculated lifetime risk ≥25%. ^ Increase of calculated risk ≥1%. ^^ Decrease of calculated risk ≤1%.

Excluding the genetic mutation carrier, 139 women were assessed, 94.2% (131/139) of whom had a mammogram available and 13% (17/131) of whom required review by a radiologist to determine BTD on the mammogram. The incorporation of BTD impacted 16.8% (22/131). When reported BTD was incorporated into the risk assessment, 6.9% (9/131) became eligible for HROBSP and MRI screening; 9.9% (13/131) became ineligible, for a net 3.1% (4/131) fewer eligible patients (Table 2, Figure 2).

**Table 2.** Eligibility according to ACR BI-RADS categories of breast tissue density.

| Density Category | Total Women with Mammograms (% of Total) | Became Eligible (% of Women with Density Category) * | Became Ineligible (% of Women with Density Category) |
|---|---|---|---|
| Density A | 10 (7.6%) | 0 (0%) | 2 (20.0%) |
| Density B | 40 (30.5%) | 0 (0%) | 6 (15.0%) |
| Density C | 58 (44.3%) | 5 (8.6%) | 5 (8.6%) |
| Density D | 23 (17.7%) | 4 (17.4%) | 0 (0%) |
| Total | 131 | 9 (6.9%) | 13 (9.9%) |

* Eligibility determined if calculated lifetime risk ≥25%.

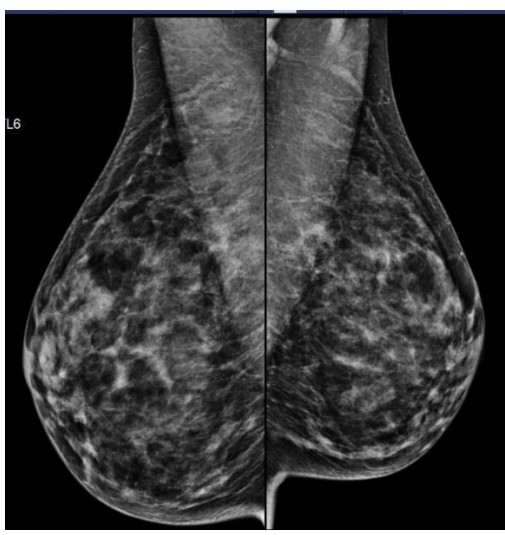

**Figure 2.** 42-year-old woman with a strong family history of BC shown to have BI-RADS category C breast tissue on screening mammograms. When BTD was incorporated into IBISv8 calculation, the lifetime risk increased from 24.2% to 32.4%, and she became eligible for HROBSP screening with B-MRI.

When comparing the number of patients who were eligible with reported BTD versus the number of patients who were eligible without including density, no significant difference was identified ($p = 0.3938$); even when considering the McNemar's exact test to account for the relatively small sample size ($p = 0.5235$). Based on these results, the proportion of eligible individuals was not significantly different between the two assessment methods.

As predicted, the 10 women who became eligible had the highest BTD as measured by BI-RADS (C or D) while 61.5% (8/13) who became ineligible had non-dense breasts (A or B). Five women who had never had a mammogram were 40–49 years old. Three women who had no mammogram available were 50–69 years and had it done outside of Ontario or remotely.

### 3.2. Feasibility of Incorporating BTD into the IBISv8 Calculation

When it became routine practice for the HROBSP NN to include the mammogram report in the referral to Genetics, it took no extra time for the GC to add BTD (BI-RADS A, B, C or D) with the other risk factors and family history information routinely collected from the patient in the risk calculation. This occurred in 87% (114/131) of assessed patients. For 13% (17/131) patients that required contact with the radiologist to help obtain the BI-RADS density score, both the GC/nurse navigator and radiologist estimated that it took about 5 min each, or 10 min in addition to the assessment, because the images were readily available for viewing. In total, for 17 patients at 10 min each, took 170 min.

## 4. Discussion

Our results demonstrated that it was feasible to incorporate BTD in high-risk assessment for BC. Including BTD in risk assessment optimized supplemental high-risk screening. No patients with non-dense tissue became eligible and 16% (8/50) ineligible, while 12% (10/81) with dense BTD became eligible, only 6% (5/81) (category C) ineligible. More patients with dense BTD were eligible while fewer with non-dense BTD required B-MRI, with an overall reduction of 3.1% (4/131) HROBSP eligible women. Although including density impacted eligibility for 17% women, the overall number of eligible women with vs. without density was not significantly different ($p = 0.3938$). Despite having been referred for a high-risk assessment for BC, 5.8% (8/139) women, and the *BRCA* carrier, had never undergone mammography.

It is well known that sensitivity of mammography is reduced in women with dense breasts, decreasing as BTD increases, 81–93% for fatty (A), 84–90% for scattered fibroglandular densities (B), 69–81% for heterogeneously dense (C) and 57–71% for extremely dense (D) breasts in women 40–74 years of age [14]. The interval cancer rate (BC detected after a normal screening study) is significantly higher in women with the most dense breasts when screened every 2 years versus every year [15]. B-MRI screening every 3–4 years in addition to mammography is cost-effective in average risk women who have the most dense breasts [16]. For women at high-risk, B-MRI is essential to permit early stage detection of BC and to reduce BC mortality. Our study found that more women with dense BTD became eligible for B-MRI while more with non-dense became ineligible, with no overall impact on use of B-MRI. Because of the masking effect of BTD, where BCs are obscured by dense tissue on mammography, contrast-enhanced B-MRI is required for early detection of BC in women with dense BTD [17].

Initial IBIS models did not incorporate BTD (v7,2004) but were updated in 2018 (v8), when BTD was shown to be more accurate in long-term assessment of BC risk [12]. Recently, researchers showed that risk stratification is improved when adding volumetric and visual mammographic density [18]. Destounis found a significantly higher proportion of high-risk women (defined as LTR $\geq$20%) when incorporating BI-RADS into IBISv8 compared with v7 (11.4% vs. 8.3% $p < 0.001$) [19]. In this study, fewer women with non-dense BTD were included in the high-risk category. A case–control study in 2019 of 474 patient participants and 2243 healthy control participants) of women aged 40–79 years found more women were included in the high-risk category, using IBISv8 instead of IBISv7 (7.1% vs. 4.8%) [20]. The

Brentnall study defined high risk as 8% 10-year or >20% lifetime risk while our study used the OBSP definition of high-risk as ≥25%. It would have been interesting to compare the impact using a similar risk assessment of ≥20%. In their study, BI-RADS assessment was a better predictor of risk than volumetric assessment of mammographic density. Our study however was a prospective evaluation of women already referred for high-risk assessment. IBISv8 is used by the HROBSP but the BTD information had not been widely incorporated as it was not being provided to the GCs performing the assessment. We showed that the HROBSP NN could provide available mammogram reports with the referral for HROBSP genetics' assessment. This occurred with 94.2% (130/138) patients assessed, which allowed for easy incorporation of BTD. Only 13.1% (17/130) of women had a prior mammogram but the report was unavailable to the GC and required radiologist input. In this instance, in took about 5 min for the GC and the radiologist to converse with each other to obtain this information.

Recently, as of October 2021, OBSP has made it mandatory for all mammograms reported through the OBSP to include the BTD according to BI-RADS categories. However, at present there is no mandatory requirement for BTD to be reported in diagnostic mammograms or screening mammogram done outside the OBSP program. BTD reporting should be mandatory for all mammograms done in Canada. In future, given these mandatory reporting requirements, it is expected that input from the radiologist will further decrease over time. In this prospective study, 16.9% (22/130) of women were clinically impacted by the incorporation of BTD with 4 fewer women requiring B-MRI. These results were consistent with the retrospective data (14.4% impacted and 4% fewer eligible women).

It is notable that 8 women from our prospective study and 8 from the retrospective data had never had a mammogram. In Ontario, population screening for BC with biennial mammograms begins at age 50. Women eligible for HROBSP assessment have a strong family history of breast and/or ovarian cancer and it is generally recommended that these women start annual mammograms (outside of OBSP) beginning at age 40 or 5–10 years prior to the earliest BC diagnosis in the family [21]. It should be noted that women considered at high risk are not included in the 2018 Canadian Task Force Preventive Health Care Guidelines [22]. Requiring a baseline mammogram prior to acceptance for HROBSP assessment would help to educate primary care providers and allow all women the opportunity to have BTD incorporated into their HROBSP assessment.

We recognize several limitations in our study. Inter-observer variation is well known in visual categorizing of BTD, and automated BTD was not used in our study. The 2013 BI-RADS 5th ed. density classification is based on the masking effect of BTD, while the 2003 BI-RADS 4th ed. [23] was based on visually estimated percentage BTD; both are associated with high rates of inter- and intra-observer variability [24,25]. Future considerations for OBSP include whether women's BC risk needs to be reassessed at some point, for example after menopause when many women's BTD decreases. There are also many women aged 30–39 y who were referred for HROBSP assessment to our centre. Neither IBISv8 or CanRisk are validated for inclusion of BTD for this age group and it is unknown whether they would benefit from reassessment after age 40. In May 2021, the BOADICEA tool was incorporated into the CanRisk tool which can now be used for unaffected women and includes similar personal risk factors as IBISv8 including BTD. The CanRisk tool provides some advantages to the genetic clinic assessment. For example, It allows for incorporation affected relatives with pancreatic and prostate cancer in addition to breast and ovarian and provides a mutation carrier likelihood for five BC risk genes (*BRCA1*, *BRCA2*, *PALB2*, *ATM*, *CHEK2*), and three ovarian cancer risk genes (*RAD51C*, *RAD51D*, *BRIP1*) compared to *BRCA1* and *BRCA2* only with IBISv8 ([26]). Unlike IBIS, CanRisk, does not however, allow for incorporation of benign breast disease such as atypical ductal hyperplasia and lobular carcinoma in situ, which are significant risk factors for BC. Which tool should be used for calculating HROBSP for unaffected women in our population is an ongoing question. Our study is limited by the small sample size but comparison with our retrospective data showed consistent results that are also in keeping with the recent larger data sets from

Brentnall and Destounis' studies [19,20]. Lastly, our study was limited by the lack of an outcomes audit and information about the stage of breast cancer in women previously assessed for eligibility in HROBSP. Including outcome information might prompt further evaluation of the eligibility threshold.

## 5. Conclusions

Our results provide support for inclusion of BTD into IBISv8 tool for purposes of HROBSP assessment of unaffected women, aged 40–69 y as it impacted the eligibility for MRI in 17% of women. The overall number of patients eligible for B-MRI was not significantly different when density was included, implying that there would not be a substantial impact to the resource requirements of the HR-OBSP. Family physicians are instrumental in assessing patient's risk for BC. If aware of a family history of BC in a woman 40 years or older, family physicians should order a mammogram to obtain BTD, as incorporating BTD will help to refine and improve risk assessment.

**Author Contributions:** Conceptualization, J.M.S., E.T., E.C., S.P. and A.R.; methodology, A.R., J.M.S., S.P., E.A. and E.T.; software, S.R., R.F. and K.R.; validation, E.A., V.H., E.S., K.A., S.Z., S.M., and C.A.; formal analysis, N.I., J.M.S. and R.F.; investigation, S.R., K.R., V.H., E.S., K.A., S.Z., C.A., S.M., S.P., J.M.S., E.T. and A.R.; resources, E.A. and J.M.S.; data curation, R.F. and J.M.S.; writing—original draft preparation, J.M.S.; writing—review and editing, A.R., E.T., S.P., E.C., V.H., E.S., K.A., S.Z., S.M., and C.A.; visualization, J.M.S. and A.R.; supervision, E.A., E.T and A.R.; project administration, E.A.; funding acquisition, E.A. All authors have read and agreed to the published version of the manuscript.

**Funding:** This research received no external funding.

**Institutional Review Board Statement:** Ethical review and approval were waived for this study due to Ottawa Hospital Research Institute approved as QI project.

**Informed Consent Statement:** Patient consent was waived due to approved QI project.

**Data Availability Statement:** Data Availability upon request in Dataverse.

**Conflicts of Interest:** The authors declare no conflict of interest.

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
