# Peer review of "Feasibility Study and Clinical Impact of Incorporating Breast Tissue Density in High-Risk Breast Cancer Screening Assessment"

_curroncol, doi:10.3390/curroncol29110688_

Round 1

Reviewer 1 Report

Review

Feasibility Study and Clinical Impact of Incorporating Breast Tissue Density in High-Risk Breast Cancer Screening Assessment

Overall, the study is valuable as a quality improvement for the high-risk screening program in Ontario, Canada. I do not know if it would be of value to other high-risk programs, that set different thresholds for inclusion.

Suggestions for edits underlined:

The High-risk Ontario Breast Screening Program (HROBSP) screens women aged 30-69 determined to be at ≥ 25% lifetime risk (LTR) with annual mammography and breast MRI. Genetic mutation carriers are automatically included, but they sought to assess the feasibility and impact of incorporating breast tissue density (BTD) into the risk assessment of women referred to HROBSP who were not genetic mutation carriers. 

The Tyrer-Cuzick model version 8 (IBISv8) incorporates BTD (for woman age 40+) and other personal risk factors along with family history of breast and ovarian cancer to determine the risk for women who have not had breast cancer. In this study, the genetic counsellors calculated LTR without and with (for those who had previously had a mammogram) inclusion of BTD. Adding breast density in IBISv8 made some women eligible who previously weren’t, all of whom were density categories C or D, but also made some ineligible who previously were, most of whom were categories A and B. There was a net reduction of 3.1% (4/131) eligible patients, which was not statistically significant. The authors refer to the “impact” as being the number of eligible women.

Feasibility referred mainly to the added time required to determine density for women whose mammogram reports did not include density information. The authors correctly point out that this time requirement will likely decrease and disappear as reporting throughout North America (albeit not worldwide) more consistently follows BIRADS, and density is always included in reports. They indicate that density reporting became mandatory in October 2021 for mammograms done through the Ontario Breast Screening program (OBSP), but this should extend to all mammograms in the province/country, and it would be ideal for them to make that recommendation in the discussion, since it would reduce the need for GC/radiologist time for women whose mammograms were done elsewhere. Since they recommend that women 40+ with a strong family history of breast cancer be referred for a mammogram outside of OBSP for the purpose of determining breast density, it goes without saying that non-OBSP mammogram reports should include breast density.  

The authors state that unlike Brentnall et al (reference 15), where the proportion of high-risk patients was higher when density was included, this study showed a net decrease in eligible patients. They should point out that the definition of high risk in this paper was 25% lifetime risk, and the Brentnall paper used 8% 10-year risk. The Ontario high-risk program has set the eligibility threshold at ≥ 25% LTR. It would be interesting to know what the outcome would be if they used 8% 10-year risk, or 20% LTR, as some centers do.  

The paper also highlights the education gap for family physicians and other referring health care providers, since these women were all referred to the high-risk program, yet several, including one with a known BRCA mutation, had never had a mammogram. International guidelines recommend risk assessment sometime from age 25-30, and clearly this is not being done consistently among their referring clinicians. Its importance should be stressed. Making it mandatory for all referrals to HROBSP to have a mammogram included in the referral would also reduce the wait for assessment if density is to be included with IBISv8.

Finally, like so many guidelines, this does not include the recommendation for an outcomes audit. Because it is known that density is only one of many variables that increase risk, it would be interesting to know the outcomes (ie stage of cancer at time of diagnosis) of the women previously assessed as eligible, who were assessed including density as ineligible. Otherwise, it is not justified to conclude that “Breast tissue density (BTD) determined by mammography better stratified risk determination of eligibility in a high-risk program.” That should be removed from the Key Points.Outcomes information might prompt a reassessment of eligibility threshold. This should be discussed/ included as a limitation.

If space is an issue, the 4 examples in Fig. 1 could be abbreviated into one image showing, for example, just a cc or mlo view of each category.

Author Response

We thank the reviewers for their helpful and insightful comments.

 Reviewer 1:

Overall, the study is valuable as a quality improvement for the high-risk screening program in Ontario, Canada. I do not know if it would be of value to other high-risk programs, that set different thresholds for inclusion.

Suggestions for edits underlined:

The High-risk Ontario Breast Screening Program (HROBSP) screens women aged 30-69 determined to be at ≥ 25% lifetime risk (LTR) with annual mammography and breast MRI. Genetic mutation carriers are automatically included, but they sought to assess the feasibility and impact of incorporating breast tissue density (BTD) into the risk assessment of women referred to HROBSP who were not genetic mutation carriers.

The Tyrer-Cuzick model version 8 (IBISv8) incorporates BTD (for woman age 40+) and other personal risk factors along with family history of breast and ovarian cancer to determine the risk for women who have not had breast cancer. In this study, the genetic counsellors calculated LTR without and with (for those who had previously had a mammogram) inclusion of BTD. Adding breast density in IBISv8 made some women eligible who previously weren’t, all of whom were density categories C or D, but also made some ineligible who previously were, most of whom were categories A and B. There was a net reduction of 3.1% (4/131) eligible patients, which was not statistically significant. The authors refer to the “impact” as being the number of eligible women.

Feasibility referred mainly to the added time required to determine density for women whose mammogram reports did not include density information. The authors correctly point out that this time requirement will likely decrease and disappear as reporting throughout North America (albeit not worldwide) more consistently follows BIRADS, and density is always included in reports. They indicate that density reporting became mandatory in October 2021 for mammograms done through the Ontario Breast Screening program (OBSP), but this should extend to all mammograms in the province/country, and it would be ideal for them to make that recommendation in the discussion, since it would reduce the need for GC/radiologist time for women whose mammograms were done elsewhere. Since they recommend that women 40+ with a strong family history of breast cancer be referred for a mammogram outside of OBSP for the purpose of determining breast density, it goes without saying that non-OBSP mammogram reports should include breast density. 

We agreed with the reviewer’s assessment of the feasibility of this study and have added their recommendation to clarify that all mammograms in the country, including screening and diagnostic mammograms, must report breast tissue density. This would reduce the time for GC/radiologist input to stratify risk assessment.

We added this point to the discussion:

“However, at present there is no mandatory requirement for BTD to be reported in diagnostic mammograms or screening mammogram done outside the OBSP program. BTD reporting should be mandatory for all mammograms done in Canada.”

The authors state that unlike Brentnall et al (reference 15), where the proportion of high-risk patients was higher when density was included, this study showed a net decrease in eligible patients. They should point out that the definition of high risk in this paper was 25% lifetime risk, and the Brentnall paper used 8% 10-year risk. The Ontario high-risk program has set the eligibility threshold at ≥ 25% LTR. It would be interesting to know what the outcome would be if they used 8% 10-year risk, or 20% LTR, as some centers do. 

We agreed with this point and added this to the discussion.

“The Brentnall study defined high risk as 8% 10-year or >20% lifetime risk while our study used the OBSP definition of high-risk as 25%. It would have been interesting to compare the impact using a similar risk assessment of 20%.”

The paper also highlights the education gap for family physicians and other referring health care providers, since these women were all referred to the high-risk program, yet several, including one with a known BRCA mutation, had never had a mammogram. International guidelines recommend risk assessment sometime from age 25-30, and clearly this is not being done consistently among their referring clinicians. Its importance should be stressed. Making it mandatory for all referrals to HROBSP to have a mammogram included in the referral would also reduce the wait for assessment if density is to be included with IBISv8.

We agreed with the suggestion and modified the 1st and 2nd key points to the following:

  1. Breast cancer risk assessment should be performed for all women ≥25-30 years of age.
  2. Women considered at high risk of breast cancer 40 years old must undergo screening mammography prior to referral to improve risk assessment.

In addition, this point and two references have been added to the introduction: In order to maximally benefit from early detection of breast cancer, international guidelines recommend that risk assessment for all women begin at 25-30 years of age[8, 9].

  1. Daly MB, Pal T, Berry MP, et al. Genetic/Familial High-Risk Assessment: Breast, Ovarian, and Pancreatic, Version 2.2021, NCCN Clinical Practice Guidelines in Oncology. J Natl Compr Canc Netw 2021; 19:77-102
  2. Monticciolo DL, Malak SF, Friedewald SM, et al. Breast Cancer Screening Recommendations Inclusive of All Women at Average Risk: Update from the ACR and Society of Breast Imaging. J Am Coll Radiol 2021; 18:1280-1288

Finally, like so many guidelines, this does not include the recommendation for an outcomes audit. Because it is known that density is only one of many variables that increase risk, it would be interesting to know the outcomes (ie stage of cancer at time of diagnosis) of the women previously assessed as eligible, who were assessed including density as ineligible. Otherwise, it is not justified to conclude that “Breast tissue density (BTD) determined by mammography better stratified risk determination of eligibility in a high-risk program.” That should be removed from the Key Points. Outcomes information might prompt a reassessment of eligibility threshold. This should be discussed/ included as a limitation.

We agreed with this point about the lack of outcomes’ audit and stage of cancer within the program. We’ve added this as a limitation in the discussion.

“Lastly, our study was limited by the lack of an outcomes audit and information about the stage of breast cancer in women previously assessed for eligibility in HROBSP. Including outcome information might prompt further evaluation of the eligibility threshold.“

We also modified the 2nd key point as noted above.

If space is an issue, the 4 examples in Fig. 1 could be abbreviated into one image showing, for example, just a cc or mlo view of each category.

We agreed and could modify the figures if required.

Reviewer 2 Report

It is a very important study. Including the assessment of breast density in the risk assessment of women in the screening program.

However, as stated by the authors , the study suffers from limitations , small sample size. and visual assessment of mammographic breast density.

Author Response

Reviewer 2:

It is a very important study. Including the assessment of breast density in the risk assessment of women in the screening program.

However, as stated by the authors , the study suffers from limitations , small sample size. and visual assessment of mammographic breast density.

We thank this reviewer for their kind comments.

Reviewer 3 Report

Dear Authors:

The manuscript "Feasibility study and clinical impact of incorporating breast tissue density in high-risk breast cancer screening assessment" by Rusnak et al has provided support for inclusion of BTD into IBISv8 tool for purposes of HROBSP assessment of unaffected women, aged 40-69y as it impacted the eligibility for MRI in 17% of women. I have just a few suggestions.

1.  Some references or information are missing.

In introduction, please add more background information about breast cancer, which can emphasize the importance of your research about the role of BTD in breast cancer. (please cite: 1. Advances in the Prevention and Treatment of Obesity-Driven Effects in Breast Cancers. Front Oncol. 2022 Jun 22;12:820968. doi: 10.3389/fonc.2022.820968. PMID: 35814391; PMCID: PMC9258420.

2. An Epigenetic Role of Mitochondria in Cancer. Cells. 2022 Aug 13;11(16):2518. doi: 10.3390/cells11162518. PMID: 36010594; PMCID: PMC9406960.

3. Mitochondrial mutations and mitoepigenetics: Focus on regulation of oxidative stress-induced responses in breast cancers. Semin Cancer Biol. 2022 Aug;83:556-569. doi: 10.1016/j.semcancer.2020.09.012. Epub 2020 Oct 6. Erratum in: Semin Cancer Biol. 2022 Jul 16;: PMID: 33035656.)

Best,

Author Response

Reviewer 3:

In introduction, please add more background information about breast cancer, which can emphasize the importance of your research about the role of BTD in breast cancer. 

(please cite: 1. Advances in the Prevention and Treatment of Obesity-Driven Effects in Breast Cancers. Front Oncol. 2022 Jun 22;12:820968. doi: 10.3389/fonc.2022.820968. PMID: 35814391; PMCID: PMC9258420.

  1. An Epigenetic Role of Mitochondria in Cancer. Cells. 2022 Aug 13;11(16):2518. doi: 10.3390/cells11162518. PMID: 36010594; PMCID: PMC9406960.
  2. Mitochondrial mutations and mitoepigenetics: Focus on regulation of oxidative stress-induced responses in breast cancers. Semin Cancer Biol. 2022 Aug;83:556-569. doi: 10.1016/j.semcancer.2020.09.012. Epub 2020 Oct 6. Erratum in: Semin Cancer Biol. 2022 Jul 16;: PMID: 33035656.)

We thank this reviewer for their suggestion of adding background information about added risk factors for breast cancer and have changed the introduction to include these references and information.

Mammographically dense breasts are very common and may contribute more cancer risk than other stronger but less common risk factors[3] including obesity [5] and mitochondrial mutations[6, 7].

Round 2

Reviewer 1 Report

Figure 2 doesn't add anything. Could be deleted (at editor's discretion).

Reviewer 3 Report

Strongly suggest for publication.